# The Purinergic P2X7 Receptor-NLRP3 Inflammasome Pathway: A New Target in Alcoholic Liver Disease?

**DOI:** 10.3390/ijms22042139

**Published:** 2021-02-21

**Authors:** Brendan Le Daré, Pierre-Jean Ferron, Thomas Gicquel

**Affiliations:** 1NuMeCan Institute (Nutrition, Metabolisms and Cancer), INSERM, INRAE, CHU—University Rennes, PREVITOX Network, F-35000 Rennes, France; brendan.le.dare@chu-rennes.fr (B.L.D.); pierre-jean.ferron@inserm.fr (P.-J.F.); 2Forensic and Toxicology Laboratory, Rennes University Hospital, 2 rue Henri Le Guilloux, F-35033 Rennes, France

**Keywords:** alcoholic-related liver disease, NLRP3 inflammasome, purinergic receptor, macrophage, P2X7R, Kupffer cell, interleukin-1β

## Abstract

The World Health Organization has estimated that approximately 3 million deaths are attributable to alcohol consumption each year. Alcohol consumption is notably associated with the development and/or progression of many non-communicable inflammatory diseases—particularly in the liver. Although these alcoholic liver diseases were initially thought to be caused by the toxicity of ethanol on hepatocytes, the latest research indicates Kupffer cells (the liver macrophages) are at the heart of this “inflammatory shift”. Purinergic signaling (notably through P2X7 receptors and the NLRP3 inflammasome) by Kupffer cells appears to be a decisive factor in the pathophysiology of alcoholic liver disease. Hence, the modulation of purinergic signaling might represent a new means of treating alcoholic liver disease. Here, we review current knowledge on the pathophysiology of alcoholic liver diseases and therapeutic perspectives for targeting these inflammatory pathways.

## 1. Overview

In its “Report on the World Alcohol and Health Situation”, the World Health Organization estimated that approximately 3 million deaths (mainly among men) were attributable to alcohol consumption in 2016—corresponding to one death every 10 s [1]. According to the literature, 28.7% of these deaths might be attributable to non-communicable diseases, including cardiovascular disease, certain cancers, neuropsychic disorders, and alcoholic liver disease (ALD) [2,3]. ALD is a leading cause of liver-related death. The physiopathology of ALD includes steatosis (fatty acid deposition in hepatocytes), steatohepatitis (inflammatory damage to the liver), and fibrosis/cirrhosis (excessive deposition of extracellular matrix), which result from both the toxicity of ethanol metabolism and complex immune reactions [4]. Here, we review current knowledge on the pathophysiology of ALD and therapeutic perspectives for targeting certain inflammatory pathways.

## 2. The Architecture of the Liver

Due to its particular position, the liver is continuously exposed (through the portal vein) to antigens from food, the intestinal flora, potentially pathogenic microorganisms, and other xenobiotics (Figure 1a). Consequently, it is the site of complex immune mechanisms that maintain the immune tolerance of intestinal antigens and enable the deployment of effective responses against pathogens [5]. The liver is mainly composed of hepatocytes (60%) but also includes endothelial cells, Kupffer cells (intrahepatic macrophages), biliary epithelium cells (cholangiocytes), stellate cells (Ito cells), dendritic cells, and intrahepatic lymphocytes. Within this structure, the innate immune system has a predominant role in orchestrating the first steps in the immune response (Figure 1b). During chronic alcohol consumption, damage to the architecture of the liver is closely linked to the toxic metabolism of this alcohol [6]. Hence, understanding this architecture is essential for characterizing the mechanisms of ALD.

## 3. The Toxicokinetics of Ethanol

Although ethanol is still used to treat some conditions (such as methanol or ethylene glycol poisoning, neurolysis, and alcohol withdrawal syndrome) and as an antiseptic, these uses have become less frequent over time as our knowledge of the compound’s toxicity and metabolism has improved [7].

The rate of alcohol absorption depends on several parameters. It is fastest when the stomach is empty and the alcohol concentration is between 20 and 30%. Hence, sherry-type alcohols (20% alcohol) will be absorbed more quickly than beer (3–8%) or spirits (40%); the latter slow down gastric emptying and inhibit the absorption of ethanol. Food (especially carbohydrates) also slows down the absorption of alcohol; the blood concentration is four-fold lower than for an empty stomach [8]. There are also sex differences; even after adjustment for body weight, women have higher blood alcohol levels than men. This might be because (i) alcohol penetrates poorly into fat and (ii) women have a higher proportion of subcutaneous adipose tissue and a lower total blood volume [8]. After the intestinal absorption of ethanol, this amphiphilic, low-molecular-weight compound diffuses throughout the body and therefore has many sites of action. Although most organs are exposed to an alcohol concentration similar to that found in plasma, the liver is an exception; the blood in the portal vein comes from the stomach and the intestines and so contains a higher concentration of ethanol.

Hepatic alcohol dehydrogenase (ADH) is the main enzyme responsible for phase I oxidation of ethanol, along with the production of acetaldehyde (the main toxic metabolite of ethanol) and nicotinamide adenine dinucleotide (NADH) [9]. The acetaldehyde is then oxidized to acetate by aldehyde dehydrogenase (ALDH). However, acetate is not the final metabolite because it can be transformed into CO_2_, fatty acids, ketones, cholesterol, or steroids [9]. Interestingly, women also have lower gastric levels of ADH in their stomachs, which reduces the metabolism of alcohol prior to intestinal absorption [8].

The effects of ethanol intolerance (such as vasodilatation—responsible in particular for facial flushing—and nausea) have been attributed to acetaldehyde. The balance between the different isoforms of ADH and ALDH regulates the concentration of acetaldehyde and is a risk factor in the development of alcoholism [10]. Genetic variants of ADH and ALDH have these characteristics and thus influence the metabolism of ethanol [11]. Certain ADH genotypes have therefore been linked to differences in alcohol consumption; people expressing the ADH2*2 allele (coding for a highly active enzyme) have a lower risk of ALD [12]. Furthermore, between 15 and 40% of people in southeast Asia have inactive ALDH (corresponding to the ALDH2*2 allele) and acetaldehyde levels that are 5 to 20 times higher than in people carrying the active isoform [9]. The accumulation of acetaldehyde in these populations may thus discourage the consumption of large quantities of alcohol in everyday life and therefore protect against alcoholism relative to Caucasian populations.

A secondary ethanol metabolism pathway involves a P450 cytochrome pathway, namely CYP2E1. This ethanol-induced oxidative pathway is mainly located in the hepatocyte’s endoplasmic reticulum and allows the production of acetaldehyde from ethanol and then acetate from acetaldehyde. Hence, ethanol’s induction of its own metabolism is thought to have a major role in ethanol tolerance among chronic consumers [9,13] (Figure 2).

Other minor pathways for ethanol metabolism include glucuroconjugation (giving rise to ethylglucuronide), sulfoconjugation (producing ethyl sulfate), the fatty acid ethyl ester synthase pathway (forming fatty acid ethyl esters), and the phospholipase D pathway (producing phosphatidylethanol) [9].

Alcohol is eliminated by several organs, including the kidneys, skin, and lungs. As is the case for absorption, many genetic and environmental factors influence the elimination of alcohol; interindividual differences of a factor of three or four are observed. In 2010, a study of 48 individuals aged 50 to 59 estimated the alcohol elimination rate at between 10 and 35 mg/100 mL blood/h [14].

## 4. Alcoholic Liver Disease

ALD occurs in three phases: alcoholic liver steatosis, alcoholic steatohepatitis, and alcoholic fibrosis/cirrhosis (Figure 3).

Alcoholic liver steatosis is characterized by the accumulation of lipids in and around the hepatocytes. It constitutes the most common and earliest form of liver damage, affecting more than 90% of individuals who consume four to five standard units of alcohol daily [15] and binge drinkers (the consumption of four to five units of alcohol in less than 2 h). Liver steatosis is reversible and therefore has a good prognosis, although chronic steatosis constitutes a risk factor for fibrotic liver diseases [16] (Figure 3). The pathophysiology of alcoholic liver steatosis is closely related to the oxidative metabolism of ethanol; the high resulting levels of NADH and acetaldehyde alter the cell’s redox balance. This induces expression of early growth response protein-1, which in turn leads to activation of the transcription factor sterol regulatory element binding protein-1C, which induces the expression of lipogenesis genes, and, on the other hand, to the synthesis of TNF-α, a lipogenic cytokine [4]. Parallel to this increase in intrahepatic lipid production, ethanol alters the synthesis of lysosomes, which are essential for the proper functioning of lipophagy [17]. Furthermore, the generation of NADH during alcohol oxidation and the mitochondrial toxicity of acetaldehyde inhibit β-oxidation, resulting in greater lipid accumulation. Ethanol has also been shown to inhibit the production of very-low-density lipoprotein, which is responsible for exporting triglycerides and cholesterol from the liver [18]. Lastly, by causing lipolysis in adipocytes, chronic alcohol consumption also increases the fraction of free fatty acids that can be captured by the liver [19]. It is also noteworthy that the early production of reactive oxygen species (ROS) linked to alcohol metabolism is responsible for a rapid increase in hepatocyte membrane fluidity. This leads to a rise in cytoplasmic, low-molecular-weight iron content and thus an increase in ROS production. This phenomenon then induces lipid peroxidation and triggers apoptosis [20]. The massive ROS production linked to the increase in membrane fluidity accounts for the perverse effects of chronic alcohol consumption. It is now clear that the microsomal ethanol-oxidizing system (MEOS) is involved in Kupffer cell activation, highlighting a key step in the transition from alcoholic fatty liver to alcoholic steatohepatitis [21,22,23]. Key components of MEOS are several forms of cytochrome P450 (CYP), especially its CYP 2E1 isoform, the NADPH-dependent cytochrome P450 reductase, and phospholipids [21]. Due to its radical scavenging properties, ethanol combines with a small fraction of hydroxyl radicals and undergoes oxidation while the remaining radicals attack phospholipids of liver cell membranes [22]. Chronic alcohol consumption upregulates MEOS activity and CYP 2E1 gene expression, leading to increased rates of alcohol degradation to acetaldehyde via MEOS and high amounts of toxic radicals that partially escape the scavenging properties of ethanol and cause liver injury [21].

Alcoholic steatohepatitis develops from underlying steatosis, occurs in 30 to 40% of individuals reporting chronic alcohol consumption, and is associated with a high short-term mortality rate [4]. In fact, lipids that have accumulated in hepatocytes are more prone to lipid peroxidation and subsequent oxidative damage. The liver’s resident macrophages (Kupffer cells) are thought to have a major role in these pathways. Kupffer cells are usually located in the lumen of the hepatic sinusoids, where they are involved in tissue repair, activation of the inflammatory response to danger signals, phagocytosis of pathogens from the portal system, angiogenesis, and the resolution of inflammation [24]. During chronic exposure to ethanol, complex interactions involving the effects of acetaldehyde, ROS, intestinal lesions mediated by endotoxins like lipopolysaccharide (LPS) and other PAMPs, and stress in the endoplasmic reticulum activate these macrophages [25] (see Section 5). An innate immune response is then initiated, leading to an increase in phagocyte activity, the recruitment of immune cells (e.g., neutrophils), the production of pro-inflammatory cytokines (such as tumor necrosis factor alfa (TNF-α), interleukin (IL)-6, IL-1α, and IL-1β), and a decrease in the production of anti-inflammatory cytokines (such as IL-10) [26] (Figure 3).

The third phase of ALD (fibrosis/cirrhosis) reflects the progression of inflammatory steatohepatitis, in which hepatocyte regeneration is severely compromised [4]. At this stage, hepatic stellate cells have a key role in the deposition of extracellular matrix—a characteristic component of fibrosis. The stellate cells are normally quiescent in the Disse space but are activated in a complex process following liver damage and the release of pro-inflammatory cytokines by Kupffer cells [4]. The progression of fibrosis during ethanol-induced chronic inflammation leads to the progressive replacement of the hepatic parenchyma by scar tissue, which compromises the liver’s metabolic and homeostatic functions and results in cirrhosis [4] (Figure 3). This cirrhotic stage is followed by severe complications, including portal hypertension and hepatocellular carcinoma (the second leading cause of liver-related cancer death) [27].

## 5. Focus on Alcoholic Steatohepatitis: The Involvement of P2X7R-NLRP3 Signaling

The pro-inflammatory cytokine IL-1β is particularly involved in the development of steatohepatitis. Following its release by Kupffer cells, IL-1β is responsible for the initiation and amplification of many immune and inflammatory responses via the synthesis of chemokines and adhesion molecules by endothelial cells and C-reactive protein by hepatocytes [28]. However, IL-1β release is dependent on activation of the NLRP3 inflammasome, an intracellular signaling complex [29].

### 5.1. The NLRP3 Inflammasome

Inflammasomes are intracytoplasmic pattern recognition receptors that are part of the innate immune system. They recognize pathogen-associated and danger-associated molecular patterns (PAMPs and DAMPs) [29]. Inflammasomes are composed of three main elements: a sensor protein, an apoptosis-associated speck-like protein containing a caspase-recruitment domain (ASC), and a pro-inflammatory caspase (caspase-1) [30]. In response to DAMPs and PAMPs, the assembly of inflammasomes activates caspase-1, which can then cleave inactive pro-IL-1β and pro-IL-18 into active IL-1β and IL-18, respectively. This inflammasome hyperactivation can also trigger pyroptosis, a cell death mechanism that slows down the replication of intracellular pathogens [31]. The nucleotide-binding domain and leucine-rich repeat containing receptors (NLRs) constitute a family of inflammasome-forming sensor proteins [32]. Twenty-two NLR-encoding genes have been identified, classified, and named according to their N-terminal domains. All NLRs (except for NLRP10) contain a central NACHT domain and a C-terminal leucine-rich repeat domain. The N-terminal domain can be a caspase activation and recruitment domain (CARD, giving rise to the NLRC family) or a pyrin domain (PYD, forming the NLRP family). The CARDs allow the direct recruitment of caspase-1, whereas the PYDs require an ASC adaptor protein [33]. Along with the NLRs, the AIM2-like and the RIG-I-like receptor families can induce inflammasome assembly [34,35].

The NLRP3 protein complex is the most thoroughly studied inflammasome. Upon activation, the NLRP3 inflammasome oligomerizes via interactions through NACHT domains; this presents the NLRP3′s PYD domain to the PYD domain in the ASC protein [36]. The latter allows the recruitment and autocatalysis of pro-caspase-1 via interaction between the respective CARD domains. This oligomerization forms a high-molecular-weight intracytoplasmic platform [37]. The cysteine protease caspase-1 cleaves inactive pro-IL-1β into active IL-1β, which can then be released into the extracellular space [38,39]. The cytokine pro-IL-18 is also reportedly cleaved by caspase-1 [40]. For activity in macrophages, the NLRP3 inflammasome requires two signals.

The first signal (known as pre-activation or priming) induces the expression of the inactive pro-IL-1β and the NLRP3 protein. Certain agonists of Toll-like receptors (TLRs, innate immunity receptors present on the plasma membrane and endosomes) are known to generate this first signal. These include LPS (an essential component of the outer membrane in Gram-negative bacteria) and certain pro-inflammatory cytokines (such as TNF-α) [29].

The second signal enables assembly of the inflammasome complex, which then converts inactive pro-IL-1β into active IL-1β. This signal can be activated by a variety of factors, including PAMPs; bacterial toxins [41,42,43]; viruses [44]; fungi [45]; protozoa [46]; DAMPs like nigericin, crystals of cholesterol, silica, or uric acid; and ATP [47]. The diversity of these signals means that they are unlikely to all activate the NLRP3 inflammasome directly; the current opinion is that these various signals converge on a common signaling pathway. Three main models have thus been suggested for activation of the NLRP3 inflammasome: the lysosomal rupture model, the ROS model, and the ion flow model.

In the lysosomal rupture model, the phagocytosis of large particles or crystals (such as uric acid, alum, silica, or β-amyloid) is responsible for ROS production, potassium efflux, phagosome destabilization, and lysosome rupture [48,49]. The release of lysosome contents (e.g., proteases like cathepsin B) into the cytoplasm is associated with caspase-1 activation [50].

In the ROS model, oxidative stress activates the NLRP3 inflammasome [30]. One study suggested that elevated ROS levels cause thioredoxin-interacting protein (TXNIP) to bind to, and thus activate, the NLRP3 inflammasome [51]. Since the main source of ROS is the mitochondrion, other studies have reported that oxidized mitochondrial DNA released during mitochondrial dysfunction can directly activate the NLRP3 inflammasome [52,53]. Furthermore, mitochondrial cardiolipin (a phospholipid from the inner wall of the mitochondrial membrane) has been shown to activate the NLRP3 inflammasome [54]. However, it is not yet known whether the ROS-dependent pathway is a major or minor contributor to inflammasome activation.

In the ion flow model, variations in the concentrations of various cations (K^+^, Ca²^+^, and H^+^) activate the NLRP3 inflammasome. Several factors have been shown to induce powerful ion fluxes. In particular, extracellular ATP (notably resulting from ethanol-induced hepatocyte death) activates the P2X7 purinergic receptors, causes a potassium efflux, and thus activates the NLRP3 inflammasome [48,55].

### 5.2. P2X7 Purinergic Receptors

Purinergic receptors are cell membrane receptors for extracellular nucleotides. They are present in all cell types, although each receptor subtype has a preferential localization [56]. The receptors are classified into two families on the basis of genetic and pharmacologic criteria: P1 receptors are selective for adenosine, and P2 receptors are activated by nucleotides (mainly ADP and ATP). The P2 class is subdivided into P2Y metabotropic G-protein-coupled receptors and P2X ionotropic channel receptors [57]. There are seven subtypes of P2X receptors (P2X1–7), for which ATP is the only physiological ligand [57]. The subtypes range from 379 to 595 amino acids in length and share 35–54% sequence homology [58]. Functional P2X channel receptors are homo- or heteromultimers formed by the association of at least three subunits [59]. Activation thus requires the binding of at least three ATP molecules to the extracellular part of the receptors [60].

The P2X7 receptor (P2X7R) is the best characterized P2X receptor with regard to its role in regulating innate and adaptive immune responses. Consistently, P2X7R is one of the most highly expressed P2X receptors in macrophages and is therefore an obvious therapeutic target [61]. The binding of ATP to purinergic receptors might stimulate a rapid potassium efflux, the production of ROS, and thus activation of the NLRP3 inflammasome in particular [62,63]. The P2X7R can also activate many different intracellular kinase or phospholipase signaling pathways [64,65,66]. Current literature data suggest that the P2X7R-NLRP3 signaling pathway is involved in many inflammatory diseases, including diabetes, gout, and pulmonary fibrosis [67].

The liver synthesizes most of the body’s nucleotides (ATP, ADP, UTP, and UDP). These key signaling molecules are recognized by hepatocytes and influence metabolic processes like biliary secretion and carbohydrate metabolism. Nucleotide levels are closely regulated, in particular, by ectonucleotidases (ecto-ADPases, ecto-ATPases, etc.) [68]. Unsurprisingly, perturbed nucleotide signaling causes inflammation, vascular damage, and the impairment of liver regeneration processes [68]. It has been reported that ethanol and fructose induce adenosine secretion, which influences lipid metabolism and promotes hepatic steatosis [68].

### 5.3. Effects of Ethanol on the P2X7R-NLRP3 Pathway

The P2X7R-NLRP3 pathway appears to be significantly involved in the pathophysiology of alcoholic steatohepatitis. Indeed, exposure to ethanol causes the passage of various PAMPS from the intestinal tract to the bloodstream, including bacterial and fungal products. LPS levels in systemic and portal blood are thus significantly increased in patients and mice with chronic alcohol consumption [69,70]. LPS interacts with its receptor, TLR4, to activate signal transduction and generate inflammatory cytokines, including TNF-α and IL-1β [71]. Peptidoglycan, a component of Gram-positive bacteria, is also detected in human ALD patients [72]. In addition, injected peptidoglycan has been shown to deteriorate liver injury and inflammation in alcohol-fed mice in a TLR2-dependent manner [73]. Particularly, N-acetylglucosamine, which is generated during peptidoglycan degradation in mice macrophages, can trigger activation of the NLRP3 inflammasome by inhibiting the function of the cytosolic glycolytic enzyme hexokinase [74]. Besides TLR4, many other pathways and receptors could play a role in activation of the NLRP3 inflammasome. For example, acute alcohol binge drinking results in increased bacterial 16S ribosomal DNA in correlation with serum LPS levels in healthy human volunteers [75]. Interestingly, bacterial DNA is recognized by TLR9 and can sensitize the liver to LPS [76]. It has been reported that chronic alcohol consumption increased gut fungal populations in mice, and the subsequent translocation of fungal β-glucans induced liver inflammation [77]. In particular, β-glucans are recognized by membrane-associated dectin-1 and cytoplasmic NLRP3 inflammasomes, resulting in IL-1β gene transcription and IL-1β secretion in human macrophages, respectively [78]. In addition, ethanol exposure activates complements, interacts with the C3aR and C5aR macrophage receptors, and leads to an increase in the production of pro-IL-1β and other pro-inflammatory cytokines [79]. This complement-dependent phase in response to ethanol occurs early in the progression of injury, since markers of complement activation are detected in macrophages within four days of ethanol feeding to mice [79].

At the same time, hepatocyte damage and the release of associated danger signals (such as ATP) linked to the oxidative metabolism of ethanol might be responsible for activation of the NLRP3 inflammasome via purinergic receptors [24]. The subsequent massive release of pro-inflammatory cytokines (particularly TNF-α) is directly responsible for hepatocyte death by apoptosis and the maintenance of alcoholic steatohepatitis [80] (Figure 4). Furthermore, ROS generated through P2X7R activation and ethanol oxidation might be responsible for NLRP3 inflammasome activation. More precisely, it was recently shown that ethanol-induced NLRP3 activation was partly triggered by downregulation of the aryl hydrocarbon receptor and upregulation of TXNIP—both of which are mediated by oxidative stress in human macrophages [81] (Figure 4). Furthermore, Heo et al. (2019) showed that ethanol decreased the expression of miR-148a in mice hepatocytes, which promoted activation of the NLRP3 inflammasome and subsequent pyroptosis of these cells [82]. Interestingly, the miRNA profile and PCR analyses also showed substantial decrease of miR-148a in the livers of patients with alcoholic hepatitis [82].

Interestingly, ethanol itself inhibits ASC phosphorylation and thereby inhibits the NLRP3 inflammasome during acute exposure in mice and human macrophages [83,84,85]. In chronic exposure, however, high levels of ROS and acetaldehyde have been reported to activate the NLRP3 inflammasome and generate pro-inflammatory effects in mice macrophages [86]. Furthermore, it was recently reported that ethanol induces the expression of P2X7R in human macrophages, which makes the macrophage more sensitive to danger signals like ATP [85].

## 6. Conclusions and Therapeutic Perspectives

In light of the above data, ethanol’s immunomodulatory effects appear to be determinants in the compound’s toxicity. This realization opens up many opportunities for understanding the mechanisms of ethanol toxicity. Kupffer cells have a central role in the pathophysiology of inflammatory ALD and are therefore attracting attention as possible therapeutic targets. In particular, the discovery of the P2X7R-NLRP3 pathway’s involvement in the pathophysiology of ALD has led to the development of new drug candidates. Molecular design studies have shown that benzene sulfonamide analogs are likely to inhibit the NLRP3 inflammasome and have prompted their development as potential drugs [87]. Furthermore, Choudhury et al. (2020) have suggested that HSP90 (a chaperone protein involved in the activation of the NLRP3 inflammasome) is a potential therapeutic target in ALD [88]. Li et al. (2018) showed that inhibition of the P2X7R-NLRP3 axis by *Pleurotus citrinopileatus* extracts decreased steatohepatitis in male C57BL/6 mice fed an ethanol-containing Lieber-DeCarli liquid diet [89]. Lastly, Freire et al. (2019) showed that the P2X7R antagonist A804598 was able to decrease liver inflammation in C57BL/6J mice fed a high-fat diet with chronic, intragastric ethanol administration [90].

In the liver, hepatocytes, stellate cells, cholangiocytes, and Kupffer cells express purinergic receptors on their plasma membrane; modulation of the latter might be valuable, especially in pathological situations [91]. Furthermore, purinergic receptors are known to be involved in alcoholic liver steatosis and hepatic fibrosis [92,93,94]. Interestingly, the P2XR antagonist pyridoxal-phosphate-6-azophenyl-2’,4’-disulfonate inhibited stellate cell proliferation and prevented the transition from alcoholic steatohepatitis to hepatic fibrosis in a rat model [95]. Zhang et al. (2018) reported that dihydroquercetin (the most abundant dihydroflavone found in onions, milk thistle, and Douglas fir bark) was able to inhibit lipogenesis in hepatic HepG2 cells by decreasing the expression of P2X7R and NLRP3 [96]. Xiao et al. (2014) showed that inhibition of hepatic TXNIP by *Lycium barbarum* polysaccharide contributed to the reduction of cellular apoptosis, oxidative stress, and NLRP3 inflammasome-mediated inflammation [97]. Lastly, Leucodin (a sesquiterpene extracted from *Artemisia capillaris*) reduced hepatic lipid accumulation by inhibiting the P2X7R-NLRP3 pathway [98]. Taken as a whole, these data suggest that the modulation of purinergic signaling might constitute a novel treatment approach for ALD.

In a broader perspective, emerging evidence shows the involvement of the P2X7R-NLRP3 inflammasome pathway in the induction of non-alcoholic fatty liver disease (NAFLD) and liver fibrosis, suggesting possible therapeutic strategies targeting the P2X7 receptor/NLRP3 inflammasome [99]. In a CCL4-induced, nonhuman primate model of liver fibrosis, treatment with a P2RX7 inhibitor (SGM-1019) improved histological characteristics of non-alcoholic steatohepatitis (NASH), protecting from liver inflammation and fibrosis [100]. Furthermore, Tung et al. (2015) found that a P2X7R blockade (using brilliant blue G and oxidized ATP) ameliorates liver fibrosis, mesenteric angiogenesis, and severity of portal-systemic shunting and improves the portal-systemic collateral vascular responsiveness in ATP in rats with bile duct ligation-induced cirrhosis, suggesting the potential of purinergic receptor antagonism in controlling liver cirrhosis-related complications [101]. MCC950, an NLRP3 selective inhibitor, was also found to partly reverse liver inflammation, particularly in obese, diabetic mice used as NASH models. In addition, such reduction of liver inflammation in NASH achieved with MCC950 partly reversed liver scarring in methionine/choline deficient-fed mice, the process that links NASH to the development of cirrhosis [102]. These recent advances in our understanding of Kupffer cell regulation may bring new hope for the therapeutic manipulation of Kupffer cells to help resolve inflammation and improve wound healing in liver disease, whether or not of alcoholic origin.

## Figures and Tables

**Figure 1 ijms-22-02139-f001:**
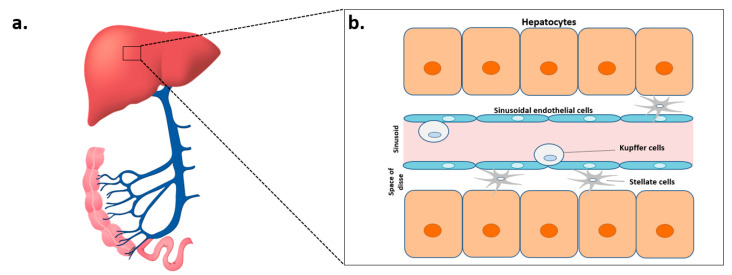
The architecture of the liver. (**a**) Representation of the hepatic portal system, showing the intestines (in pink), the portal vein (in blue), and the liver (red). (**b**) Representation of the main cell types in the liver: hepatocytes, sinusoidal endothelial cells, Kupffer cells, and stellate cells.

**Figure 2 ijms-22-02139-f002:**
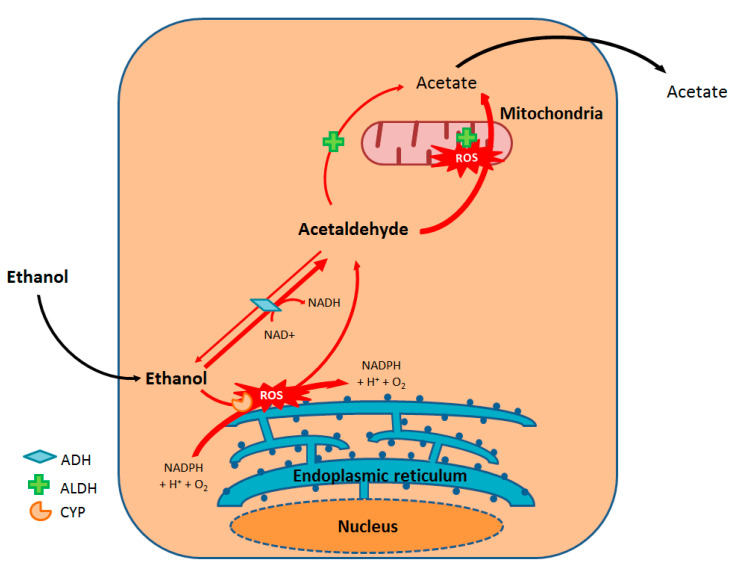
Oxidative metabolism of ethanol. ADH: Alcohol dehydrogenase; ALDH: Aldehyde dehydrogenase; CYP: Cytochrome P450.

**Figure 3 ijms-22-02139-f003:**
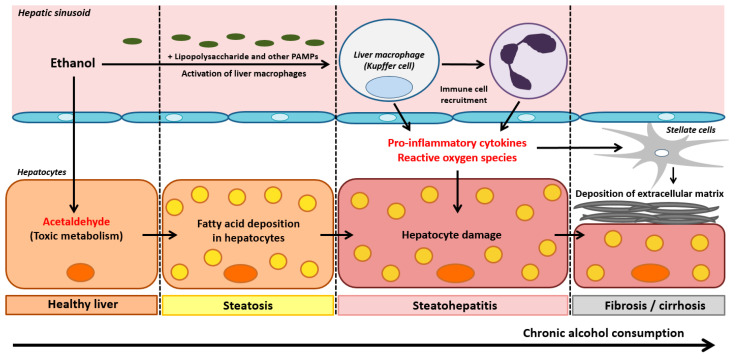
Pathophysiological mechanisms of the three stages in ALD: hepatic steatosis, steatohepatitis, and fibrosis/cirrhosis. PAMPs: Pathogen-associated molecular patterns.

**Figure 4 ijms-22-02139-f004:**
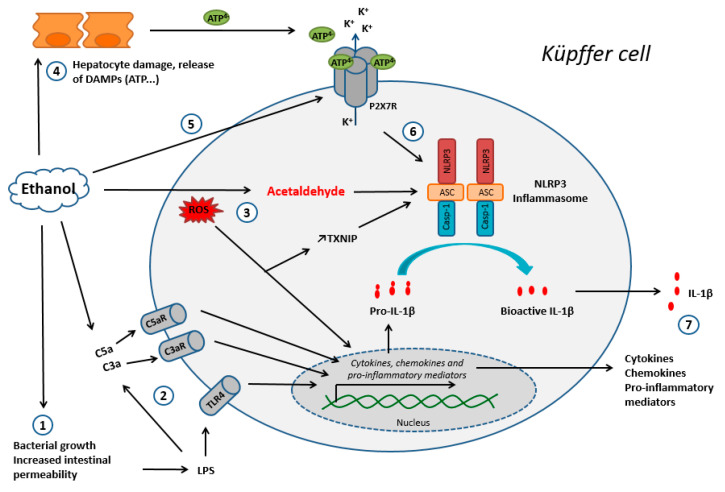
The main mechanisms of Kupffer cell activation during chronic ethanol exposure. Step 1: An increase in intestinal permeability, leading to the release of LPS into the systemic circulation and activation of the TLR4 receptor. Step 2: Activation of the complement system, in particular via LPS. Steps 1 and 2 lead to the production of chemokines, cytokines, and other pro-inflammatory mediators. Step 3: Oxidative metabolism of ethanol; ROS and acetaldehyde lead to an increase in TXNIP expression, activation of the NLRP3 inflammasome, and the production of pro-inflammatory cytokines (ii). Step 4: Hepatocyte damage due to the oxidative metabolism of ethanol results in the release of danger signals, including ATP. Step 5: Ethanol induces the expression of P2X7R, further sensitizing the cell to danger signals like ATP. Step 6: ATP activates the NLRP3 inflammasome via purinergic receptors. Step 7: Release of the chemokines, cytokines, and inflammatory mediators produced upstream. Abbreviations: DAMP: danger-associated molecular pattern; LPS: lipopolysaccharides; ROS: reactive oxygen species; TXNIP: thioredoxin interacting protein.

## Data Availability

Data sharing not applicable. No new data were created or analyzed in this study. Data sharing is not applicable to this article.

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
