# Peer review of "The Purinergic P2X7 Receptor-NLRP3 Inflammasome Pathway: A New Target in Alcoholic Liver Disease?"

_ijms, 2021, doi:10.3390/ijms22042139_

Round 1

Reviewer 1 Report

The authors reviewed the implication of inflammatory pathways in the pathophysiology of ALD. In particular, they focus on the NLRP3 inflammasome and its activation by ethanol. The more general part of this review is clearly written, gives a good overview of the current knowledge and discusses general mechanisms not necessarily related to ethanol or its indirect consequences. However, by focusing only on PSX7, a relative narrow view of the concept of ethanol in the activation of the inflammasome is presented in the more specific part of the review. This part of the paper would largely benefit from a broader point of view and the authors should consider extending the review to other aspects beyond the role of the P2X7 receptor in inflammasome activation.

  1. The authors focus on the gram-neg./LPS/TLR4 pathway and its related consequences as an activator of the inflammasome in ALD. To my knowledge, there is data out in the literature potentially implicating gram-positive bacteria and their products such as peptidoglycans as well as fungi in ALD pathogenesis. They likely also interfere with the inflammasome and this should be discussed in realtionship to ethanol in the review.
  2. Besides TLR4 and P2X7, many other pathways and receptors could play a role in activation of the inflammasome. Especially, TLR2 or other intracellular TLRs might be relevant in this context. Another potential candidate could be NOD-like receptor associated inflammasome activation. Their implication in relationship with ethanol should also be more specifically discussed.
  3. In general, for clarity of the paragraphs dealing with ethanol and activation of the inflammasome, it would be useful to clarify what data are derived from in vitro and animal studies. Given the lack of good animal models in ALD research, these data cannot not always be extrapolated to the human situation. Specific care should be taken to add, if available, human data confirming the animal findings or state that human data on specific topics are not available.

Reviewer 2 Report

The purinergic signaling pathway is a highly complex system playing a major role in inflammatory processes. The role of ATP released by damaged cells in the extracellular space, the activation of the P2X7 receptor, together with the involvement of NLRP3 inflammasome and pro-inflammatory cytokines production is now well-proven in the liver. Emerging evidence show the involvement of the P2X7 receptor/NLRP3 inflammasome in the induction of NAFLD/NASH and liver fibrosis. (Rossato M et al, 2020, Mridha AR et al. 2017) In addition to NAFLD, the beneficial effects of P2X7 antagonism in rats with bile duct ligation-induced cirrhosis has also been demonstrated. (Tung HC et al. 2015) In the present study, Brendan et al. reviewed the purinergic P2X7 receptor-NLRP3 inflammasome pathway to investigate whether it is a new target in alcoholic liver disease. This is a comprehensive review for the pathogenesis of alcoholic liver disease regarding purinergic P2X7 receptor-NLRP3 inflammasome pathway. However, this pathway seems to involve in the final process of hepatic inflammation regardless of alcoholic and non-alcoholic liver disease. This issue should be addressed in this review and which can provide more border discussions of liver fibrogenesis with various etiologies. 

Round 2

Reviewer 1 Report

All my comments have been addressed and the manuscipt has been improved.

Author Response

We thank the reviewer for their comments.